# GENE FINDING REVISITED: IMPROVED ROBUSTNESS THROUGH STRUCTURED DECODING FROM LEARNING EMBEDDINGS

## ABSTRACT

Gene finding is the task of identifying the locations of coding sequences within the vast amount of genetic code contained in the genome. With an ever increasing quantity of raw genome sequences, gene finding is an important avenue towards understanding the genetic information of (novel) organisms, as well as learning shared patterns across evolutionarily diverse species. The current state of the art are graphical models usually trained per organism and requiring manually curated data sets. However, these models lack the flexibility to incorporate deep learning representation learning techniques that have in recent years been transformative in the analysis of protein sequences, and which could potentially help gene finders exploit the growing number of the sequenced genomes to expand performance across multiple organisms. Here, we propose a novel approach, combining learned embeddings of raw genetic sequences with exact decoding using a latent conditional random field. We show that the model achieves performance matching the current state of the art, while increasing training robustness, and removing the need for manually fitted length distributions. As language models for DNA improve, this paves the way for more performant cross-organism gene-finders.

## 1 INTRODUCTION

Genes are patches of deoxyribonucleic acid (DNA) in our genome that encode functional and structural products of the cell. The central dogma of biology states that these segments are transcribed into ribonucleic acid (RNA) and in many cases translated into the amino acid sequences of proteins. In recent years, the machine learning community has dedicated considerable attention specifically to studying proteins, and solving various protein-related tasks, with the aid of deep learning. This focus has resulted in impressive advances within the field (Detlefsen et al., 2022; Jumper et al., 2021; Rao et al., 2020; Shin et al., 2021). Less attention has been paid to the DNA sequences themselves, despite the fact that finding genes in a genome remains an important open problem. Due to technological advances, the rate by which genomes are sequenced is rising much more rapidly than we can reliably annotate genes experimentally, and without proper gene annotations, we lack information about the proteins encoded in these sequences. In particular, for taxonomies that are sparsely characterised or highly diverse, such as fungi, this hinders us from extracting essential information from newly sequenced genomes.

The wealth of available genomic data suggests that this is an area ripe for a high-capacity deep learning approaches that automatically detect the most salient features in the data. This potential has in recent years been clearly demonstrated in the realm of proteins where deep learning has proven extremely effective in both the supervised setting (Alley et al., 2019; Hsu et al., 2022; Jumper et al., 2021) and in the unsupervised setting (Rao et al., 2021; 2020; Vig et al., 2021). In particular, embeddings obtained in transformer based protein language models have pushed the boundaries for performance in many downstream sequence-based prediction tasks. The advantages of such models are two-fold: 1) they enable pre-training in cases where unlabelled data far outweighs labelled data and 2) they have demonstrated the ability to learn across diverse proteins. We are currently witnessing an emerging interest in language models for DNA as well, but progress in this area has proven more difficult than for its protein counterpart. In a completely unsupervised setting the amount of DNA data is orders of magnitude larger than that of proteins, and the signals are

correspondingly sparse. For instance, Eukaryotic genomes consist of millions to billions of DNA base pairs but only a small percentage are genes and an even smaller percentage codes for protein (approx. 1% in the human genome). Genes also have an intricate structure which places demands on a high degrees of consistency between output labels predicted at different positions in the sequence. In particular, genes contain both coding segments (called CDS or exon) and intron segments. Only the coding segments are retained in the final protein, while the introns are removed. The process in which introns are removed is called splicing, which occurs after the gene is transcribed from DNA to RNA. After introns are spliced out, the RNA is translated to amino acid sequences (the protein product). Each amino acid is encoded by a codon (a triplet of DNA nucleotides) in the RNA sequence. Due to this codon structure, the annotation of the coding sequences in the gene must be extremely accurate, as shifting the frame of the codon with just one will result in a nonsensical protein. Gene prediction thus consists of correctly annotating the boundaries of a gene as well as the intron splice sites (donor/acceptor sites), a task challenged both by the imbalanced data but also by the extreme accuracy needed.

The current state-of-the-art in gene-finding relies on Hidden Markov Models (HMMs) and exact decoding (e.g. Viterbi) to ensure the required consistency among predictions at different output positions. To make these methods work well in practice, considerable effort has been put into hand-coded length distributions inside the HMM transition matrix, and a careful curation of the training data to ensure that the length statistics are representative for the genome in question. The resulting HMMs have dominated the field for more than two decades. However, their performance still leaves a lot to be desired, they are generally difficult to train, and have no mechanism for incorporating learned embeddings and context dependent learned length distributions. These models can be improved by incorporating them with external hints and constructing pipelines (Hoff et al., 2016) but they are not compatible with deep learning advents that have revolutionised adjacent fields. The goal with this paper is to develop a new approach that is compatible with contemporary deep learning practices, can be trained without manual feature engineering and careful data curation, while maintaining the capability for exact decoding.

Here we present an approach, which we term GeneDecoder, to gene prediction that is able to both incorporate prior knowledge of gene structure in the form of a latent graphs in a Conditional Random Fields as well as embeddings learned directly from the DNA sequence. This approach proves easy to train naively while still achieving high performance across a range of diverse genomes. We highlight that the resulting model is very flexible and open to improvement either by including external hints or by improvement of the current DNA sequence embeddings. We benchmark against three other supervised algorithms (Augustus, Snap, GlimmerHMM) and find that the performance of our model competes with that of the state-of-the-art (Scalzitti et al., 2020) without a strong effort put into model-selection. However, as pre-trained DNA models start to emerge and improve we expect that the full potential of this approach will be realised.

## 2 RELATED WORK

Current gene prediction algorithms are Hidden Markov Models (HMM) or Generalized HMMs. These include Augustus (Stanke & Waack, 2003) , Snap (Korf, 2004)., GlimmerHMM (Majoros et al., 2004) and Genemark.hmm (Borodovsky & Lomsadze, 2011). All these models are trained fully supervised and on a per-organism basis. Genemark also exists in a version that is similarly trained on one organism but in an iterative self-supervised manner (Ter-Hovhannisyan et al., 2008). In practice, gene prediction is often done through meta-prediction pipelines such as Braker (Hoff et al., 2016), Maker2 (Holt & Yandell, 2011) and Snowyowl (Reid et al., 2014), which typically combine preexisting HMMs with external hints (e.g. protein or RNA alignments) and/or iterated training. Focusing on the individual gene predictors, Augustus is currently the tool of choice in the supervised setting, according to a recent benchmark study (Scalzitti et al., 2020). It is an HMM with explicit length distributions for introns and CDS states. The model also includes multiple types of intron and exon states emitting either fixed-length sequences or sequences from a length distribution given by a specific choice of self-transition between states. This intron model has been shown to be key to its performance; without it Augustus was found to be incapable of modelling length distributions in introns correctly (Stanke & Waack, 2003). Models like Snap and GlimmerHMM follow similar ideas, but differ in their transition structure. In particular, GlimmerHMM includes a splice site model from Genesplicer (Pertea et al., 2001). These HMM-gene predictors are known to be

Table 1: Label sets used in the CRF model. The Direction labels are used to support processing genes in both the forward (F) and reverse (R) directions. The Codon numbering is required to keep track of the reading frame. To avoid the need for users to specify these labels, we use a latent CRF which internally processes the full label set, but emits only the base labels.

| Description | Base label | Direction | Direction + Codon |
|---|---|---|---|
| Exon/CDS | $E$ | $E_F, E_R$ | $E_{1,F}, E_{2,F}, E_{3,F}, E_{1,R}, E_{2,R}, E_{3,R}$ |
| Intron | $I$ | $I_F, I_F$ | $I_{1,F}, I_{2,F}, I_{3,F}, I_{1,R}, I_{2,R}, I_{3,R}$ |
| Donor splice site | $D$ | $D_F, D_R$ | $D_{1,F}, D_{2,F}, D_{3,F}, D_{1,R}, D_{2,R}, D_{3,R}$ |
| Acceptor splice site | $A$ | $A_F, A_R$ | $A_{1,F}, A_{2,F}, A_{3,F}, A_{1,R}, A_{2,R}, A_{3,R}$ |
| Non-coding/intergenic | $NC$ | | |

sensitive to the quality of the input data set. For instance, the Augustus training guidelines specifies requirements for non-redundancy, non-overlapping genes and restrictions to single transcripts per genes. These considerations are not merely theoretical. In the preparation of the baseline results for this paper, we have attempted to retrain several of these models on our own data set splits, but were unable to obtain competitive results. The Augustus guidelines also report diminishing returns for data sets with more than 1000 genes, and highlights the importance of quality over quantity in the training set in this scenario. These recommendations are sensible in a low-data regime, but we might wish to relax them as we obtain more genomic data. It would be convenient if we could train gene predictor models on lower quality data sets, including multiple, potentially conflicting transcripts arising for instance from alternative splicing. The goal of this paper is to explore modelling strategies towards this goal.

Eventually, we would also like to train cross-genome gene predictors, for instance using featurizations obtained from pre-trained language models. Following the success of pre-trained models on protein sequences, DNA language modelling is starting to emerge as a field. Currently, two such models are available: DNABert (Ji et al., 2021) and GPN (Benegas et al., 2022). Both are trained on a single organism, each across a whole genome with a sequence window of 512 nucleotides. While DNABert is a standard transformer based model for which long sequences can be computationally prohibitive, GPN (Benegas et al., 2022) is based on dilated convolutions where it could be possible to extend the sequence length. In this work we perform preliminary explorations of using GPN embeddings for gene prediction.

## 3 METHODS

### 3.1 MODEL

**Latent conditional random fields** Genomic datasets are highly unbalanced, containing only a very small percentage of the donor and acceptor labels. Since identification of these labels is paramount for correct annotation, the model must be designed to deal robustly with this. Furthermore, we wish to encode prior knowledge of gene structure in order to produce consistent predictions. Finally, the model must be flexible in the type of input used, so that context dependent embeddings can be learned directly from the sequence. To fulfil these requirements, we choose a Linear chain Conditional Random Field (CRF) (Lafferty et al., 2001) model. The CRF can, like the HMM, encode gene structure in its graph, but because it models the conditional probability of the output labels, rather than the joint distribution of input and output, it is well suited for integration with a embedding model. In doing so, we can hope to learn a richer representation of the input, while still training the model end-to-end.

The input to the embedding model are one-hot encoded values $\mathbb{X} = \{A, C, G, T, N\}$, where N is unknown nucleotides. As output labels, we have Exon ($E$), Intron ($I$), Donor ($D$), Acceptor ($A$) and Non-coding ($NC$). The CRF learns transition potentials (i.e. unnormalized transition probabilities) between these states, but we can manually set particular of these potentials to $-\inf$ to allow only biologically meaningful transitions, thus following a similar strategy as that employed in classic HMM gene predictors. Two important restrictions are 1) directionality, ensuring that forward and reverse coded genes are not intermixed, and 2) codon-awareness, which ensures that the coding

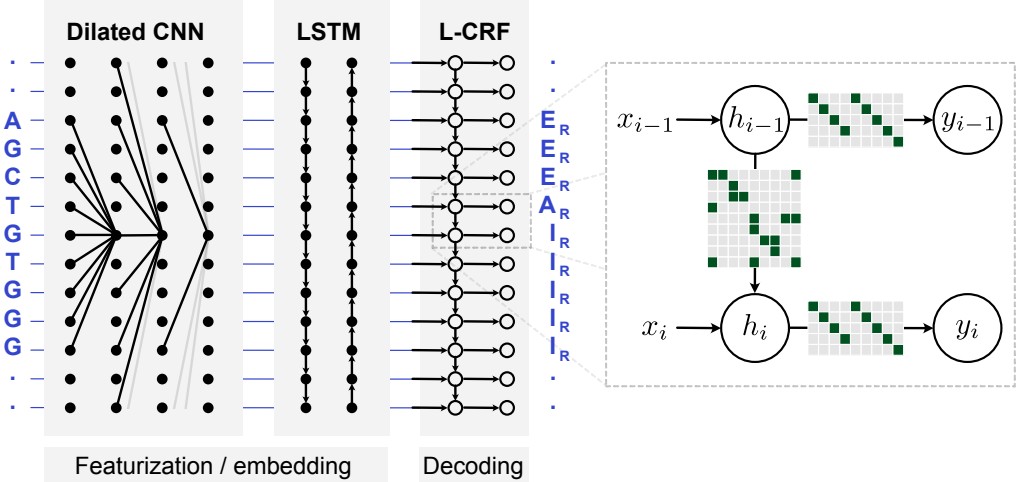

Figure 1: Overview of the model. The current best version of GeneDecoder has a feature model that processes the input first with dilated convolutional layers with residual connections and thereafter a bidirectional LSTM. The feature model outputs un-normalized label probabilities per positions which are then processed by the latent CRF to output label sequences in a manner consistent with the allowed transitions and emissions.

regions of genes are always a multiple of three. To encode this in the transition potentials, we expand the label set (Table 1). Note that in addition to the exon states, it is necessary to expand also the intron, acceptor and donor states such that codon position can be tracked through the intron regions.

The likelihood for a CRF takes the form:

$$P(\mathbf{y}|\mathbf{x},\theta) = \frac{1}{Z} \exp \sum_{n=1}^{N} \left( T_{y_{n-1},y_n} + \theta(x_n) \right) \tag{1}$$

$T$ is the learned transition weights of the observed labels and $\theta$ the learned input embedding. The partition function $Z$ is the summed score over all possible label sequences:

$$Z = \sum_{y' \in Y} \exp \sum_{n=}1^{N} \left( T_{y'_{n-1},y'_n} + \theta(x_n) \right) \tag{2}$$

In the standard CRF, the states appearing in the transition table will be observed during training. In our case, we do not wish to specify the direction and codon-index apriori. HMM-based genefinders solve this problem by formulating the Markov process in a latent discrete space, and coupling it to the output states through a learned emission probability table. A similar solution is not easily tractable for CRFs. However, as shown by Morency et al, it becomes feasible if we introduce a many-to-one deterministic mapping between latent and observed states (Morency et al., 2007). This is a natural assumption to make in our case, as illustrated by the individual rows in Table 1. The corresponding Latent CRF (L-CRF), includes a summation over the set of latent states emitting a given observed state.

$$P(\mathbf{y}|\mathbf{x},\theta) = \frac{1}{Z} \exp \sum_{n=1}^{N} \left( \sum_{h' \in \mathbb{H}_{y_{n-1}}} \sum_{h \in \mathbb{H}_{y_n}} (T_{h',h} + E_{h,y_n}) + \theta(x_n) \right) \tag{3}$$

where $\mathbb{H}_{y_n}$ denotes the set of hidden states emitting the value taken by $y_n$. The learned transition matrix $T$ is now over the hidden states and the emission matrix $E$ denotes the emission potentials from hidden to observed states. Similarly, the partition function now also sums over the set of hidden

states that emits $y$:

$$Z = \sum_{y' \in Y} \exp \sum_{n=1}^{N} \left( \sum_{h' \in \mathbb{H}_{y_{n-1}}} \sum_{h \in \mathbb{H}_{y_n}} (T_{h',h} + E_{h,y_n}) + \theta(x_n) \right) \tag{4}$$

In our case, the $\mathbb{H}_{y_n}$ matrix is given by the rows in Table 1. The non-blocked entries in the transition matrix $T$ are learned (see examples of an adjacency matrices in Figure A.2). We assume that gene structure is similar on the forward and reverse strands and therefore share the learned transition potentials between the two directions. Figure 1 shows a full overview of our model. In the remainder of the paper, we will refer to this model decoding architecture as CRF and L-CRF interchangably.

**Feature Model Embeddings** Although the embedding function $\theta$ in a CRF was traditionally based on manually engineered features, modern autograd frameworks make it possible to learn such functions using e.g. neural networks, and train both the featurization and decoder models end-to-end. This allows us to use a complex, context dependent mapping from the one-hot encoded input sequence as the conditioning input to our CRF. As a proof of concept, we here use a simple combination of a (dilated) CNN and an LSTM. The former ensures that we see beyond single nucleotides, avoiding the need to manually encode at the codon level, and the latter models the remaining sequential signal. We disregarded more expressive architectures such as transformers because we in the purely-supervised setting are in a limited data regime. However, we anticipate that pre-trained language models for DNA will eventually replace the need for learning new feature embeddings per organism (actually, one could meaningfully define it as a success criterion for DNA language models). We briefly explore this potential in the result section below.

**Training** The model were trained using AdamW on the on the Negative Log Likelihood, using a batchsize of 64. We used early stopping based on the validation set likelihood to avoid overfitting.

**Inference** Viterbi decoding is used to find the best label sequence under the model. Alternatively sampling can be performed according to the label probabilities, either across the entire sequence or for segments. Posterior marginal probabilities per position are calculated with the forward-backward algorithm and can be used to assess model certainty for specific positions.

**Choice of performance metrics** Accurately predicting the direction, as well as the donor and acceptor sites (i.e. start and end of introns), is highly important for predicting the correct protein. For this reason Matthews Correlation Coefficient as well as Macro weighted F1 score is reported a the label set containing exon/CDS, intron, donor and acceptor labels.

## 4 EXPERIMENTS

### 4.1 DATA

Sequence and general feature format (gff3) annotation files are obtained for each genome as follows: Human - Gencode, Aspergillus Nidulans and Saccharomyces Cerevisiae - EnsembleFungi, Arabidopsis Thaliana - EnsemblPlants, C. Elegans Wormbase (Frankish et al., 2021; Cunningham et al., 2022; Howe et al., 2017).

No augmentation or inspection of the data was performed, with the exception of ensuring the total length of CDS segments in each gene was divisible by 3, ensuring that valid protein product was possible. Gene annotations were extracted with a random flanks of a length between 1-2000 nucleotides. These relatively short flank lengths were chosen to decrease training time. The maximum allowed gene length was capped according to the gene length distribution in the species to ensure a Representative data set but simultaneously exclude extreme outliers in gene length that would heavily increase compute time. The data set was homology partitioned at a threshold of 80% protein sequence similarity into train, validation and test sets. The test sets were used for benchmarking. The length of the flank can influence the performance depending on the inference task. For some ablation experiments simpler feature models have been chosen in order to reduce compute, these results may therefore reflect only a relative performance rather than the full potential.

## 4.2 BENCHMARK ALGORITHMS

We benchmark our method against three other widely used gene prediction software (Augustus, GeneMark.hmm, Snap and GlimmerHMM). Given the sensitivity of these methods to the training sets, pre-trained parameter set for each species was used where available. This means we cannot rule out that sequences in our test set are part of the training set of the baseline models. Our reported results are thus a conservative estimate of our relative performance. There exists several versions of the GeneMark algorithm depending on the desired user case (Borodovsky & Lomsadze, 2011; Besemer et al., 2001; Lukashin & Borodovsky, 1998). We chose GeneMark.hmm as this is a supervised pre-trained per species algorithm, which matches the scope here.

## 4.3 BENCHMARKING ACROSS EVOLUTIONARILY DIVERSE SPECIES

To demonstrate the performance of our approach we benchmark against four widely used programs on five phylogenetically diverse well-studied species typically used in benchmarking (Table 2). Our model, GeneDecoder, was trained separately on genes from each organism, which is the standard for such gene-finders (Stanke et al., 2006; Borodovsky & Lomsadze, 2011). For the other models, pre-trained versions of the programs for the given species, were used in order to not reduce performance by re-training on a data set that was insufficiently curated for their model.

We find that only Augustus achieves similar performance to our model, with Snap performing markedly lower across all species we it tested on. GlimmerHMM and GeneMark.hmm shows better performance but it still has significantly lower predictive power compared to Augustus. This further cements previous findings that Augustus achieves state of the art performance (Scalzitti et al., 2020).

Our model, GeneDecoder competes with or outperforms Augustus, which is a state-of-the-art gene prediction software, across the benchmark, with the exception of the lower F1 score for *S. Cerevisiae*. The low F1 score for S. Cerevisiae across all algorithms is due to the low number of intron-containing genes. Less than 1% genes contain introns in the data set for this organism. Since the F1 score is highly influenced by performance on underrepresented label categories this reduced performance comes from poor prediction of introns and donor/acceptor sites. It was not possible to obtain Augustus' training data set for this species, but according to the official training instructions of the software, data sets must be curated to contain a high proportion of intron-containing genes. This would explain discrepancy in the F1 score for Augustus and GeneDecoder, however it also highlights that our model is not highly influenced by the quality of the data set. We expect that the perfomance of GeneDecoder would improve with a curated dataset or oversampling of intron containing genes.

We further compare Augustus with our model by training and testing on the original Augustus data sets for *Human* (Stanke, 2004). The training data set is in a low data regime, consisting of 1284 training genes, which is not ideal for deep learning settings. Furthermore, the Augustus data set is strictly homology reduced, allowing no more than 80% sequence similarity at the protein level. Nevertheless, our model matches the performance of Augustus on its own data set (see Figure A.5). The difference in predictive performance on non coding labels are likely due to a manually set self-transition probability of 0.99 for non coding labels in the Augustus model. This improves Augustus' performance on non coding labels but might come at the expense of predicting other labels. While improving Augustus performance requires the use of external hints chosen by the user, our model may be improved by learning better embeddings.

## 4.4 LEARNED EMBEDDINGS IMPROVE PREDICTIONS

To explore the performance contribution of different aspects of our model we performed ablations, where parts of the model were excluded or simplified part of it (Table 3). Note that these models were trained on a smaller set of the *A. thaliana* data set.

First, to test the capabilities of the CRF alone, a single linear layer is implemented to map each position from the one-hot encoded DNA sequence to the hidden states. Two graphs of different complexity were used for this experiment (see Table 1, Figure A.2 and A.2 for graphs). While both Linear-CRFs exhibit very poor predictive power, the more complex codon graph, which encodes knowledge of codon structure, clearly performs better that the simple graph, illustrating the need

Table 2: Comparison of algorithms

| Model | Human | | A. Nidulans | | A. Thalaiana | | S. Cerevisiae | | C. Elegans | |
|---|---|---|---|---|---|---|---|---|---|---|
| | MCC | F1 | MCC | F1 | MCC | F1 | MCC | F1 | MCC | F1 |
| **GeneDecoder** | **0.81** | **0.83** | **0.90** | **0.91** | **0.90** | **0.82** | **0.88** | 0.46 | **0.92** | **0.94** |
| **Augustus** | 0.78 | 0.81 | 0.87 | 0.81 | 0.87 | 0.70 | **0.88** | **0.53** | 0.76 | 0.75 |
| **Snap** | 0.36 | 0.41 | N/A | N/A | 0.54 | 0.54 | 0.85 | 0.45 | 0.55 | 0.58 |
| **GlimmerHMM** | 0.74 | 0.76 | N/A | N/A | 0.77 | 0.78 | N/A | N/A | 0.78 | 0.80 |
| **GeneMark.hmm** | 0.62 | 0.71 | N/A | N/A | 0.64 | 0.56 | N/A | N/A | 0.59 | 0.70 |

Table 3: Ablations of feature model on A. Thaliana

| FEATURE MODEL | MCC | F1 |
|---|---|---|
| Linear + CRF (simple graph) | 0.15 | 0.11 |
| Linear + CRF (codon graph) | 0.34 | 0.19 |
| LSTM | 0.79 | 0.75 |
| LSTM + CRF | 0.86 | 0.84 |
| DilCNN + LSTM + CRF (GeneDecoder) | 0.89 | 0.85 |
| GPN + CRF | 0.74 | 0.75 |

for a higher graph complexity to improve modelling performance. This is consistent with similar observations made for HMM-based gene predictors Stanke & Waack (2003).

By introducing a feature model such as an LSTM, performance is greatly enhanced. While an LSTM trained alone also displays decent performance it is evident from Figure A.5 that without the CRF, splice sites are often not labelled correctly which is vital for producing consistent predictions.

Interestingly training a linear layer on fixed embeddings from the GPN language model (Benegas et al., 2022) achieves high performance (Figure A.5). This indicates that the masked pre-training of the GPN model captures DNA sequence patterns relevant for gene prediction, despite only having trained on a single organism and primarily on non-coding DNA. The emergent field of DNA language modelling could have a significant effect on downstream DNA modelling tasks, as they have had in the protein field. These ablation studies reveal that with a good feature model, there is less need for as complex a graph structure. The feature model can instead learn complex dependencies directly from the sequence.

The best performing model, which we refer to as GeneDecoder, uses a feature model with a combination of dilated convolutions as well as a bidirectional LSTM, which we al. We hypothesise that the increased performance over a LSTM alone is due to better long range information provided by the dilated convolutions. The details of the architecture are available in Appendix A.4).

### 4.5    LEARNED EMBEDDINGS ACCURATELY MODEL LENGTH DISTRIBUTIONS

Accurately capturing length distributions of introns and exons is essential for a well performing gene predictor. Along with explicitly modelling exon length distributions, Stanke & Waack (2003) had to introduce an intron sub-model with states modelling both explicit length distributions as well as fixed length emissions, to meaningfully capture intron lengths.

We find that our model readily learns these signals directly from data (Figure (4.5)). The latent CRF without learned embedding is not sufficient to reproduce the true lengths resulting in a flattened distribution similar to Stanke & Waack (2003). However, with the learned embeddings, GeneDecoder captures the length distributions faithfully, especially for exons. It is very promising that the pre-trained embeddings GPN model is also able to capture the length distributions accurately. The GPN embeddings were not fine-tuned in this case. This further cements that incorporating deep learning, and particularly unsupervised pre-trained embeddings, is a promising direction for GeneDecoder and

genefinders in general. Especially when expanding the model to perform cross-species prediction. However, DNA language modelling is still a nascent field and there are currently many unanswered questions on how to transfer the masked modelling concept to DNA sequences as compared to proteins. These concern both the sparsity of information, the longer sequences and the much longer range signal that likely needs to be captured for high performance in many downstream tasks. The GPN model is trained on DNA segments of 512 nucleotides which might exclude capturing many longer range signals and affect performance.

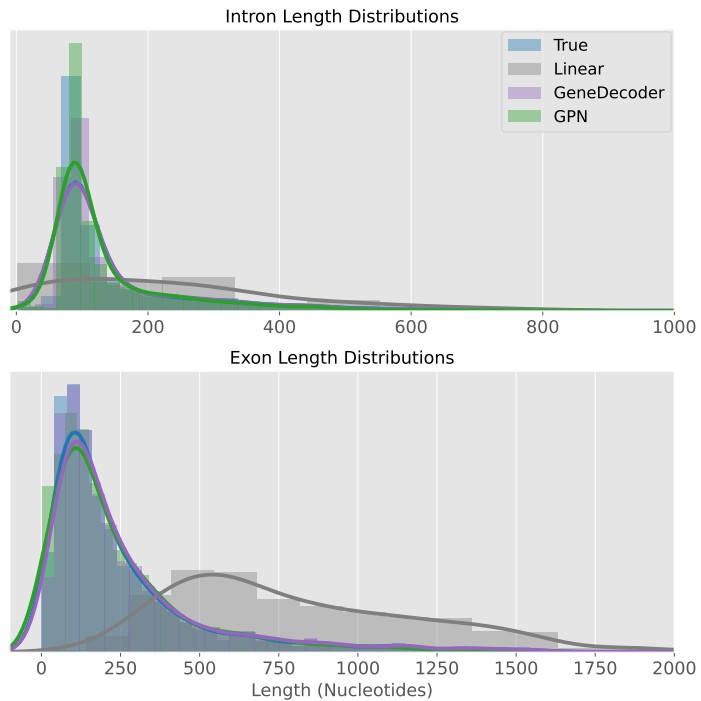

Figure 2: Learned embeddings capture length distributions of more accurately

### 4.6 LATENT STRUCTURE CAN INFER UNOBSERVED LABELS

To test the capacity of the latent graph of our model we explore whether it can learn unobserved states directly from the sequence. One such example is inferring the directionality of genes. Genes can be encoded on both strands of the double-helical DNA but usually only one strand is sequenced. Genes that lie in the "reading" direction of sequenced strand are designated as forward and genes that would be read on the opposite strand as reverse. We train our model on a directionless set of observed labels (see *simple-labels-DA* in Appendix A.1). Since the the training data does not provide any information about directionality, the model must learn this directly from the sequence to be able to distinguish between directions.

Figure 4.5 4 shows the comparison between our model trained on the directionless label set as well as a set including directional labels (see *simple-direction-DA* in Appendix A.1). Although there is a reduction in performance between the two training regiments, the model is still able to learn and infer direction. Surprisingly the reduced performance does not originate from confusion between forward and reverse labels but rather due to the model's propensity to over predict non-coding labels (see A.5).

## 5 CONCLUSION

Here we present a novel approach, GeneDecoder, to gene prediction combining the advantages of graphical models with the flexibility and potential of deep learned embeddings.

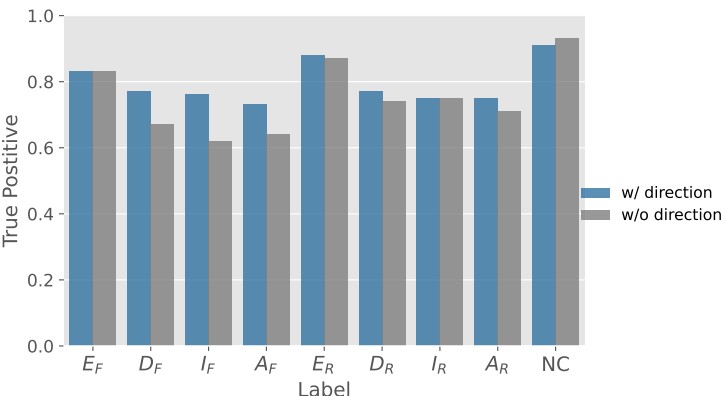

Figure 3: Inferring unobserved labels directional labels

We find that the Latent CRF gives us not only the ability to preserve the graph-encoding of prior knowledge, as in the state-of-the-art HMMs gene predictors, but also offers an advantage by allowing us to learn embeddings directly from the sequence. This advantage is not only visible in the performance, but also in the low effort required to construct data sets and train the model. The greatest advantage, however, might well be the great flexibility of the learned embeddings and the potential to improve them as the DNA modelling field advances.

Even now this seems evident from the preliminary studies performed here using embeddings from a DNA language model trained only on a single species, with very short input sequences. We expect that as the field progresses language models more specifically tailored to DNA will emerge, modelling longer range interactions and training across multiple species. Such results have already been demonstrated by the success of the Enformer model (Avsec et al., 2021), which demonstrated the importance of long range information as well as the ability to learn downstream tasks that were not trained on.

One potential limitation of our model is that the amount of non-coding DNA (in our cases the flanking regions) in the training data set can affect the performance of the model on data with significantly more non-coding DNA. This issue can be resolved in the training process by scaling the training to larger data sets containing longer non-coding regions. As this presents a technical challenge, along with a time and resource challenge, rather than a scientific one we leave it for future work. Our model provides the capacity to include external hints in the inference process. Either via sampling high probability pathways around fixed labels but also via modification of the hidden state probabilities. The latter method can even be used as a strategy during training, highlighting the flexibility of out model. Lastly, model per position certainty is directly provided model in the form of posterior marginal probabilities. We leave it for future exploration to rigorously benchmark this.

We also expect that language modelling will alleviate issues like this by modelling far better and more informative embeddings of the raw sequence. But we leave this for future work. In this regard we view the model presented here as an initial iteration that can only improve as DNA modelling does. We anticipate that future iterations of GeneDecoder will incorporate some form of self-supervised pre-training across various species which could vastly improve not only prediction tasks in species with rich gene annotation data but also in novel or understudied organisms.

## 6 CODE AVAILABILITY

Code will be available on github upon publication.

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

# A    APPENDIX

## A.1    LABEL SETS

Generally five types of labels are used:

$E$ : Exon/CDS (i.e. the coding part of a sequence)

$I$ : Intron

$D$ : Donor site (one of the two splice sites)

$A$ : Acceptor site (one of the two splice sites)

$NC$ : Non-coding or intergenic regions of the genome.

Subscript F and R (e.g. $E_F$ and $E_F$) are used to denote the forward and reverse direction of the gene. Numbered subscripts are used to denote codon structure (i.e. $E_{1,F}$ is the first exon nucleotide in a codon in the forward direction).

**Simple-DA**:

The simplest set of states used.
$$\mathbb{Y} = \Big\{ E, D, I, A, NC \Big\}$$

**Simple-direction-DA**:

Expands the set of states to include information about the direction/strand of the gene.

$$\mathbb{Y} = \Big\{ E_F, D_F, I_F, A_F, E_R, E_R, I_R, A_R, NC \Big\}$$

**Codon-direction-DA**:

Further expands the set states to include not only direction but also codon structure.

$$\mathbb{Y} = \Big\{ E_{1,F}, E_{2,F}, E_{3,F}, D_{1,F}, D_{2,F}, D_{3,F}, I_{1,F}, I_{2,F}, I_{3,F}, A_{1,F}, A_{2,F}, A_{3,F}, \\ E_{1,R}, E_{2,R}, E_{3,R}, D_{1,R}, D_{2,R}, D_{3,R}, I_{1,R}, I_{2,R}, I_{3,R}, A_{1,R}, A_{2,R}, A_{3,R}, NC \Big\}$$

## A.2    GRAPHS

Graphs are represented as adjacency matrices and are named according to the label sets described in Appendix A.1. A grey cell signifies a forbidden transition while the weights of the green cells are learned by model.

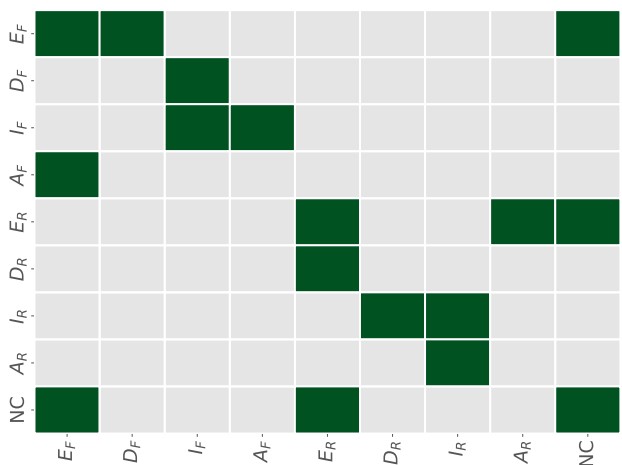

Figure 4: Graph of allowed transitions for *simple-direction-DA* labels

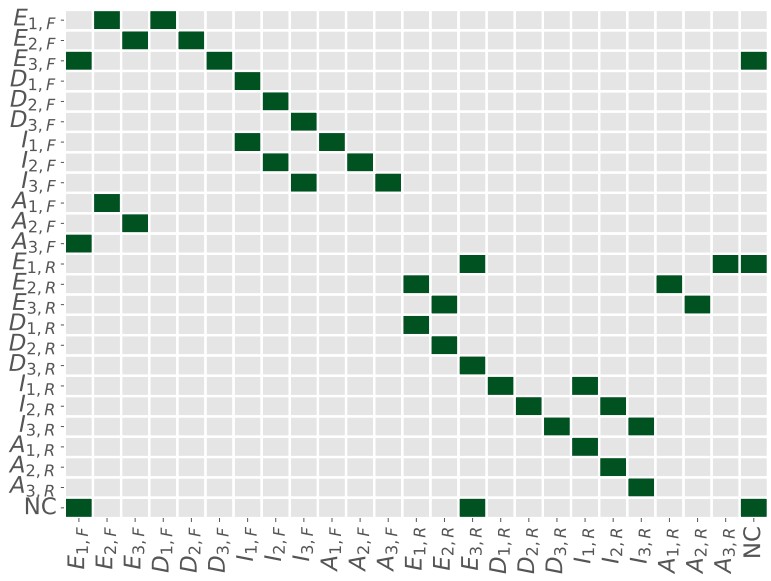

Figure 5: Graph of allowed transitions for *codon-direction-DA* labels

## A.3 EMISSIONS

A grey cell signifies a forbidden emission while the weights of the green cells are set to 1 (i.e. an allowed emissions).

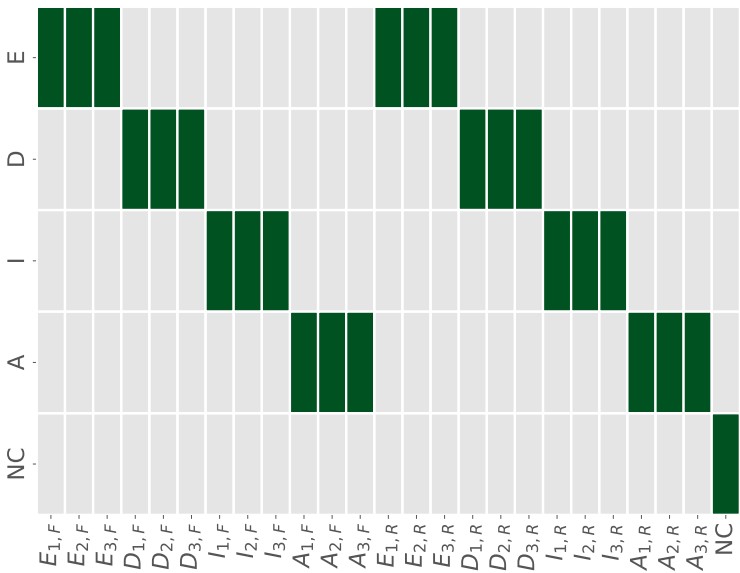

Figure 6: Allowed emissions from *codon-direction-DA* labels to *simple-DA* labels

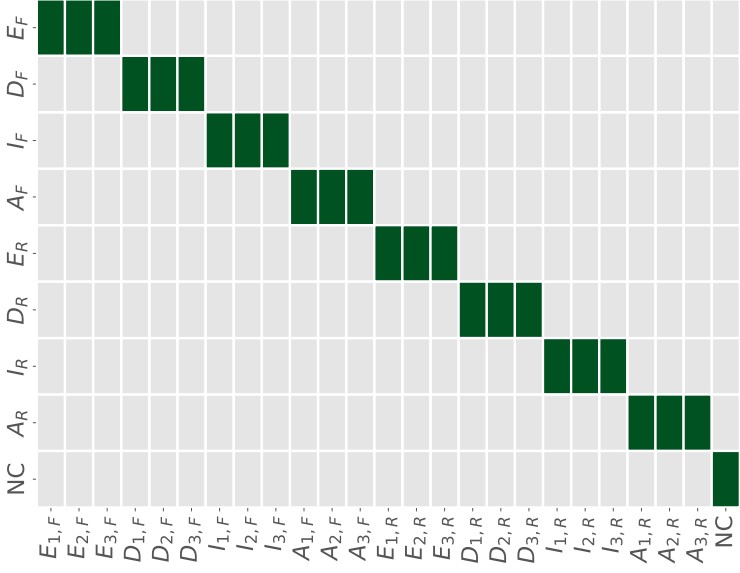

Figure 7: Allowed emissions from *codon-direction-DA* labels to *simple-direction-DA* labels

### A.4 GENEDECODER FEATURE MODEL ARCHITECHTURE

Feature Model

Where $|\mathbb{X}|$ is the number of channels in the input, and $|\mathbb{H}|$ is the size of the set of hidden states. In most cases $\mathbb{X} = \{A, C, G, T, N\}$.

DilatedCNN(

      Layer 1 : Conv1d(in channels = $|\mathbb{X}|$, out channels = 50, kernel size = (9,), stride = 1, dilation = 1), ReLU

      Layer 2 : Conv1d(in channels = 50, out channels = 50, kernel size = (9,), stride = 1, dilation = 2), ReLU

      Layer 3 : Conv1d(in channels = 50, out channels = 50, kernel size = 9,), stride = 1, dilation = 4), ReLU

      Layer 4 : Conv1d(in channels = 50, out channels = 50, kernel size = 9,), stride = 1, dilation = 8), ReLU

      Layer 5 : Conv1d(in channels = 50, out channels = 50, kernel size = 9,), stride = 1, dilation = 16), ReLU

      Layer 6 : Conv1d(in channels = 50, out channels = 50, kernel size = 9,), stride = 1, dilation = 32), ReLU

      )

LSTM(

      LSTM layer : lstm(50, hidden layers = 100, bidirectional = True)

      Output layer : linear(input size = 200, output size = $|\mathbb{H}|$, bias = True,

      dropout(0.2)), ReLU

      )

### A.5 SUPPLEMENTARY FIGURES

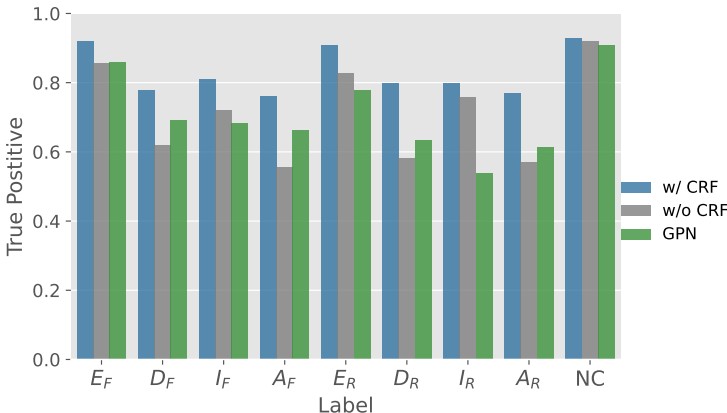

Figure 8: Comparison of true positive rate per label for a LSTM-CRF, LSTM and Linear-GPN-CRF model respectively.

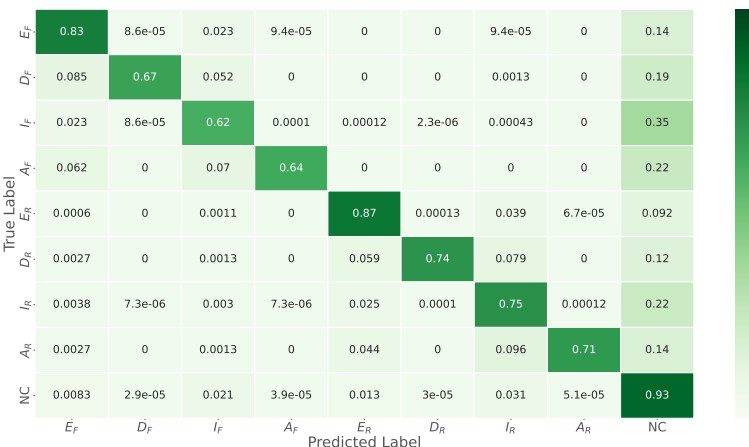

Figure 9: Confusion matrix for inference of direction on test set performance of model trained on *simple-DA* label set

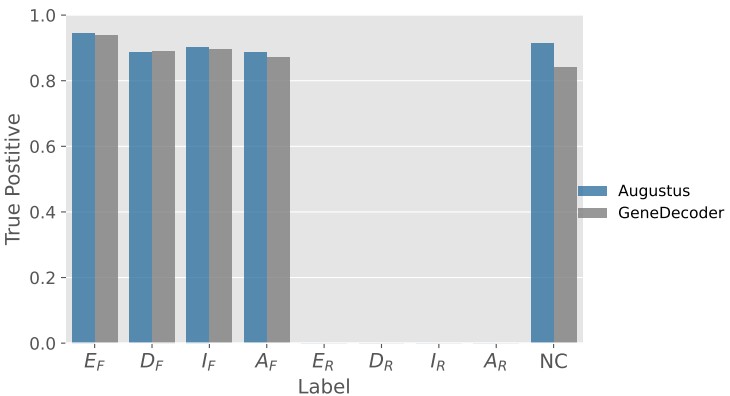

Figure 10: True positive rate per label for training and testing on the original Augustus *Human* datasets

