# OpenReview forum: "Gene finding revisited: improved robustness through structured decoding from learning embeddings"
_ICLR.cc/2023/Conference — Submitted to ICLR 2023_

### Official Review · Reviewer_6LHj · 2022-10-19

**Confidence:** 4
**Correctness:** 3
**Technical Novelty And Significance:** 2
**Empirical Novelty And Significance:** 3
**Recommendation:** 8

**Clarity, Quality, Novelty And Reproducibility:**

I think the paragraph beginning "The current state of the art" should include citations.  E.g., the first sentence of this paragraph should include citations.  And the first mention of software packages such as Glimmer should be accompanied by a citation.

In Section 4.1, there are some confusing statements about how the model is set up, where we are told, for example, that random flanks were added, but not precisely how these flank lengths were selected.  The text implies that this was handled differently in different experiments.  These choices need to be clearly described, for reproducibility. Similarly, why is the gene length capped at two different values, and which experiments are these caps used for?

I was confused by the organization of the results.  I thought Section 4.2 was going to provide an initial comparison of the proposed method to the current state of the art.  But that section simply lists the what those methods are, and then we skip immediately to the ablation experiments.  I think the ablation experiments should come after the results that show that the method works well across many species. This seems to be what appears in Section 4.6.

Minor:

"dominated the field for more than a decade" -> "... more than two decades"

"HMM model" is redundant.

The Stanke and Wacke citation used \cite{} rather than \citep{}.  This happens several other places as well.

"and highlighting" -> "and highlights"

"recommendation are" -> "recommendations are"

"identification ... are" -> "identification ... is"

Section 3.1 should clarify what the output labels are.

Throughout, there is no need to capitalize the name of a method, even if the associated abbreviation necessarily uses capitals.  E.g. "Linear chain Conditional Random Field (CRF)" -> "linear chain conditional random field (CRF)"

"choice ... depend" -> "choice ... depends"

"takes a onehot encoded sequences" -> "takes a onehot encoded sequence"

"of the CRF:" -> "of the CRF is used:"

"state:" -> "state"

"Figure 3.1" -> "Figure 1"

"to label to unnormalised label" -> ??

"on the either across" -> ??

"one the hidden" -> "on the hidden"

"asses" -> "assess"

"inference set" -> "inference"

The description of annotation sources in Section 3.1 is not clear.  Provide URLs and make clear what those citations refer to.

"A. Thaliana" -> "A. thaliana"

Section 4.3 should do a better job of explicitly using the labeling scheme from Table 1, so we know which line is being discussed at each step.

Section 4.4 mentions the name "GeneDecoder" without explaining that this is the name of the proposed method.  The name is formally introduced later.  I think you should state the name in the introduction.

"both strand" -> "both strands"

"genes that lay" -> "genes that lie"

"of sequenced" -> "of the sequenced"

"models" -> "model's"

"evolutionary" -> "evolutionarily"

"The benchmark limited and specific" -> ??

"model ," -> "model,"

"but it is still has" -> "but it still has"

"previous findings" -> What previous findings? This either requires a citation or a reference to some previous result in the current paper.

"Human" -> no italics, no capitalization

"models closely" -> "model closely"

"mebeddings" -> "embeddings"



**Strength And Weaknesses:**

The paper does a very good job of providing a concise and accessible description of the gene finding problem for an ICLR audience.

Not much novelty from a machine learning perspective.

The main missing piece in this paper, in my opinion, is an investigation into the types of errors that the system makes in comparison to other methods.  I want to know whether the system is missing some exons entirely, just getting the splice sites wrong, hallucinating new genes that don't exist, etc.

The other missing piece is an evaluation of how well the system works in practice on an actual genome, rather than on snippets of a genome selected around a gene.  As is, the paper is more like a proof of principle than a full solution.


**Summary Of The Paper:**

This manuscript describes a straightforward application of standard deep learning techniques to the important problem of gene finding in eukaryotic genomes.  It is actually quite surprising that something like this has not been done before, since this is a central problem in genomics that is still not well solved despite decades of work in the area.  State-of-the-art methods have used HMMs for more than two decades. This paper provides convincing evidence that a deep neural network approach to this problem has promise.

**Summary Of The Review:**

From a machine learning perspective, very little here is new or innovative, but that's OK when the task is simply to apply standard methods to solve an important problem. The writing and organization are clear, though the paper is poorly proofread, with many typos and grammatical errors.

---

> ### Author Response · Authors · 2022-11-19
> **First reply**
>
> We thank the reviewer for his/her thorough review and apologize for the many grammatical errors in our original submission. We have corrected them as well as improved the overall quality of the writing. It should now be more clear how the model is set up as well as how the experiments are conducted (see also the response to the other reviewers). We have also reordered the result section as suggested by the reviewer, and clarified the specification of flanking regions.
>
> The reviewer makes two excellent points about missing aspects of the paper: how well it works in practice across a larger stretch of genome and an analysis of the errors. Running on entire chromosomes at once requires involves a few technical challenges regarding stitching together featurizations which are are currently solving - but unfortunately did not finish in time for the rebuttal revision of the manuscript - we hope to be able to include this in the camera-ready version.
>
> We agree that a systematic analysis of errors would be appropriate. We have discussed this at length but some of the metrics are not trivial to define (e.g. exons are not always missing in their entirety; sometimes a splice site difference is a known isomer etc). We also looked into assessing performance at the protein level via alignment of target and predicted proteins, but also here decisions have to be made (parameters for the protein alignment algorithm etc). Due to these complexities, we did not manage to finish this study before the revision deadline. We expect to converge on these issues soon, and will include some quantification of errors in the camera ready version.

---

### Official Review · Reviewer_DUdq · 2022-10-23

**Confidence:** 3
**Clarity, Quality, Novelty And Reproducibility:** The quality is not above acceptance t…
**Correctness:** 2
**Technical Novelty And Significance:** 2
**Empirical Novelty And Significance:** 2
**Recommendation:** 3

**Strength And Weaknesses:**

The paper argues the prior models for gene prediction “lack the flexibility to incorporate deep learning representation learning techniques” as a potential drawback. This sets the motivation of the paper to develop a deep learning compatible gene prediction model. However I do not see why deep learning compatibility, per se, is a drawback of the prior works. Thus, in my opinion, the paper is poorly motivated from both the biological and methodological perspectives. The issue becomes even more problematic, as the paper demonstrates that the proposed deep learning-compatible method does not significantly improve the performance in any tangible aspect over the baseline considered. Unfortunately, all these are on top of the fact that the paper misses a key family of baseline algorithms for gene finding, i.e., GeneMark, GeneMark-S, etc. which is discussed in the paper but never compared with.

**Summary Of The Paper:**

The paper develops a new algorithm for gene finding which is compatible with embeddings obtained from deep and transfer learning.

**Summary Of The Review:**

The paper can be substantially improved by 1) better motivating the need for a new gene finding algorithm, 2) discussing the proposed method more clearly without over-referring to the Appendix, which I found highly distracting, 3) including key remaining baselines, and 4) editing the grammatical errors which occasionally harms the clarity of the message. I do not thing in the current state the paper is ready for publication.

---

> ### Author Response · Authors · 2022-11-19
> **First reply**
>
> We thank the reviewer for his comments. We agree with the reviewer that our original submission did not properly motivate our contribution. We have now reworked the paper to strengthen the motivation and vision of the paper.
>
> We agree with the reviewer that deep learning compatibility is not a goal in its own right. Our personal motivation is that we need genefinding capabilities for detecting promising enzyme candidates for protein engineering in sparsely annotated species such as fungi, and the current state-of-the-art in gene-finders is not providing sufficiently accurate results. Our hypothesis is that we should be able to better transfer gene-prediction capabilities between species if we can learn transferable features. This paper constitutes a first step in this direction - separating gene finder algorithms into a feature-extraction phase and an exact decoding phase. Our work revealed that current pre-trained language models for DNA are not yet powerful enough  to constitute a competitive alternative to the task-specific embedding that we use in this paper. But there is much activity in the DNA language modelling field, and we believe that this will happen in the near future. We also do stress that our method is currently the state-of-the-art in gene-finding - outperforming its main competitor Augustus in all but one species.
>
> It was indeed a mistake that GeneMark was not included in our comparison table. We have now included it. We chose GeneMark.hmm, which we also motivate in the paper, because it is the closest match for this setting among the different visions of Genemark.

---

### Official Review · Reviewer_ggXe · 2022-10-24

**Confidence:** 3
**Correctness:** 3
**Technical Novelty And Significance:** 3
**Empirical Novelty And Significance:** 3
**Recommendation:** 6

**Clarity, Quality, Novelty And Reproducibility:**

Clarity - Paper is very easy to read and follow

Quality - Moderate quality work - requires more analysis to justify the findings

Novelty - Moderately novel - CRFs are the mainstay of POS tagging tasks and this is the first application in the gene embedding space. Though novel, more analysis is required to ascertain the need with respect to state of the art methods.


Reproducible - The paper is easily reproducible and gives clarity.

**Strength And Weaknesses:**

Strengths

State of the art gene finding

Pioneer attempt to combine pre-trained embeddings with gene finding

Incorporation of domain knowledge to improve performance

Exhaustive experimentation

Cross organism experiment verification


Weaknesses

No clarity or intuition on why CNN + LSTM does better than pre-trained models

While an empirical ablation has been done, it seems non-intuitive that the pre-trained embeddings do not help much, considering they are tested on similar downstream non-coding tasks.

Novelty needs to be explained for architecture

For example, how does the combination of embeddings work with neural HMM based formulation work as compared to the state of the art HMM methods?

The efficiency study for these methods is missing. What do we gain by using this architecture (which takes time to compute) against others? This is relevant especially because the values of Augustus’ baseline performance is quite close.

No statistical significance values reported

**Summary Of The Paper:**

The authors have presented a novel method of gene finding - where the task is to label spans of text the functional labels.  The presented method uses a neural algorithm that leverages conditional random fields (CRF) over the state of the art HMM methods. They provide a way to fuse pre-trained representations of genomic sequences with their architecture for this task. The novelty is in terms of architecture, incorporating domain information in the form of allowed direction of state changes and experimentations across organisms.

**Summary Of The Review:**

The authors have proposed a new neural architecture, with elegant ways of including domain knowledge, for the task of gene finding. This could potentially be a new fine tuning task for gene embedding research and help practitioners in unearthing genes among unlabelled data. The experiments have been conducted well and the problem is relevant and exciting.


At this point, I would recommend a weak reject because I am not convinced this neural architecture, though it incorporates domain knowledge, is significantly better than the state of the art methods in any setting, (for example, Augustus also seems to perform well in cross organism settings).

===============================
Several of the comments have been addressed. So increasing the score.

---

> ### Author Response · Authors · 2022-11-19
> **First reply**
>
> We thank the reviewer for the comments. We acknowledge that the motivation and model description were not very clear in the original submission, and have reworked the manuscript to make these points clearer. It is an excellent point that it is surprising that the simple CNN+LSTM featurization is superior to pre-trained embeddings - which also puzzled us originally. We believe there are several potential reasons for this: 1) the embeddings have been trained only on short segments (e.g. 512 for GPN) which is likely not enough to capture long range interactions across the gene sequences (e.g. between donor and acceptor sites), 2) The embeddings have been trained using standard masked language modelling approaches, which might not induce a inductive bias conductive to gene finding, 3) in our current setting, the embedding models are not fine-tuned - we are currently exploring whether this might make a difference. Generally speaking, language modelling for DNA is not as mature as that of proteins, with only a small handful of models published so far, and results so-far indicate that NLP-like methods translate less well to DNA than they did to protein - both due to longer sequences and a sparser signal. We and others are looking into improving language models for DNA to better support gene-finding, but it is outside the scope of the current paper. We have added a brief discussion of these points to the paper.
>
> Various alternative combinations of neural networks and HMM-like models have been proposed in the literature. The earliest is perhaps the by Krogh, Riis from 1999, termed hidden neural networks, which are similar to the CRF approach used in our paper. More recently, “Deep HMMs and “Neural HMMs”, but as far as we are aware, much of this work is focused on parameterizing higher order sequential transitions using neural networks. In our case, we need the first-order Markovian assumption in order to readily impose our domain knowledge of gene structure, and the higher-order approaches are therefore not directly applicable to us. The CRF has the advantage that we need not be generative about our learned feature embeddings, and therefore appeared to us as the most natural choice. We have made this motivation of the model clearer in the new revision of the paper.
>
> Regarding efficiency, the complexity of the CRF the same as that of the HMMs used in earlier gene predictors (O(K^2N)), and the featurization does not dominate for typical sequence lengths. If anything, the size of K (the transition matrix) is smaller in our model that the Augustus model, because we do not need to allocate additional states for intron length distribution modelling.
>
> We agree that it would have been most proper to report statistical significance numbers. We ultimately decided against this because we are unable to retrain the baselines (see response to reviewer 1 above), and therefore could not establish relevant uncertainties for the methods we compared to.

---

### Official Review · Reviewer_rvN5 · 2022-10-25

**Confidence:** 5
**Clarity, Quality, Novelty And Reproducibility:** See above.
**Correctness:** 2
**Technical Novelty And Significance:** 1
**Empirical Novelty And Significance:** 3
**Recommendation:** 1

**Strength And Weaknesses:**

Overall, the problem of annotating genes is important, the method is reasonable and the manuscript is understandable. The methodological novelty is low, as the method is a combination of standard CRF, LSTM and dilated CNN models. The experimental setup is flawed and the results are poor.

I couldn't tell what the model used by GeneDecoder is. It involves a neural network, whose architecture is described in A.4. However it also uses a Latent CRF. I think the NN outputs a representation which forms the input to the CRF? This isn't actually stated.

Benchmarking gene prediction tools is challenging because many of the annotations within gene databases (e.g. GENCODE) are derived from computational predictors. Thus it can be hard to tell whether a predictor is good at discovering real biology or simply recapitulating the errors made by previous predictors. There exists a benchmark for this task, G3PO (Scalzitti et al 2020), cited by the authors.  Unfortunately, the authors did not use this benchmark and instead used an ad-hoc strategy with many issues (see below).

The authors compared to pretrained versions of existing models, so differences in performance could result from differing training sets. When the authors compared against Augustus using a training set similar to Augustus's, results were similar or worse.

The authors trained and tested on genes of the same species. This not a proper simulation of the target application, in which a new species is sequenced and its genome must be annotated from scratch.

In splitting genes into train and test sets, the authors ensure that all isoforms of the same gene are placed in the same set. However, since many genes overlap, the same sequences likely appear in both train and test sets. This pitfall is described in detail in the following paper. A better strategy would be to split train and test by chromosome (or by species; see previous note).
https://pubmed.ncbi.nlm.nih.gov/34837041/

**Summary Of The Paper:**

The authors tackle the problem of annotating genes in newly-sequenced genomes. They develop a model called GeneDecoder which uses a combination of a CRF, LSTM and dilated convolutional layers. The authors show that the model relearns several properties of genes, including directionality and length distribution. The model achieves similar predictive performance to existing method Augustus.

**Summary Of The Review:**

See above.

---

> ### Author Response · Authors · 2022-11-19
> **First reply to rebuttal**
>
> We thank the reviewer for his comments. In the revised manuscript, we have rewritten the methods section of the manuscript and modified Figure 1 to improve the clarity, and hope that the reviewer will find the connection between various subcomponents more clearly explained now.
>
> We agree with the reviewer’s point that it can be difficult to trust the annotations in databases. For certain species, GENCODE allows us to filter for annotation quality - and in our experiments we have thus filtered for  annotations by those supported by experimental evidence as well as manual curation whenever possible. The G3PO is certainly an important resource, but it was unfortunately not directly suitable for our purposes since it contained very few sequences from each species (e.g ~20 protein coding genes for human).
>
> We fully agree that it would be preferable if we could retrain benchmark algorithms such as Augustus so that we can compare on exactly the same datasets. However, Augustus proved difficult to train reliably in our hands, and despite many attempts, we were unable to retrain Augustus to the same performance levels as reported in the literature. To make sure we did not represent it unfairly, we therefore chose to use a pre-trained version. We cannot rule out that some of our test sequences appear as training sequences in the original Augustus training, and our expectation is therefore that this comparison gives an advantage to Augustus, and thus constitute a conservative estimate of our performance. We note that also G3PO uses pretrained versions of the algorithms. Regarding the comparison to the state-of-the-art, we are uncertain why the reviewer reached the conclusion that we are similar or worse than Augustus. As shown in Table 2, we outperform Augustus on all species, except on S. Cerevisiae, which is due to a very low occurrence of introns in this genome (which we discuss in the paper).
>
> We strongly agree that the ultimate goal of gene predictors is prediction on completely novel species. To our knowledge, however, no current method meets this standard: Augustus, GeneMark, Snap and GlimmerHMM are all trained and used on a per species basis. The self supervised version of Genemark which can be trained on novel genomes is usually also trained per species.
> We acknowledge that the splitting procedure was perhaps not strict enough in our original submission. In the revised manuscript, we now split the train, test and val partitions according to protein homology, with a threshold of 80% sequence similarity. This is to our knowledge a standard approach (e.g. mentioned in the augustus training guide: https://vcru.wisc.edu/simonlab/bioinformatics/programs/augustus/docs/tutorial2015/training.html).
>
> The reviewer makes a good suggestion of splitting according to chromosomes as well. We are currently running these experiments but unfortunately, they did not complete in time to be included in the revised manuscript for the rebuttal. We will include these results in the camera-ready version.

---

> > ### Comment · Reviewer_rvN5 · 2022-11-20
> > **2022-11-20**
> >
> > Thank you for the response. These comments move the paper in the right direction, but many issues remain. My score is unchanged.

---

### Decision · Program_Chairs · 2023-01-20

**Decision:**

Reject

**Justification For Why Not Higher Score:**

While the reviewers have more polarized scores, overall all reviewers noted there's limited novelty from technique and machine learning perspective and it's also lacking the level of extensive evaluation, testing, that several reviewers mentioned. The authors should be commended for revising and replying to the comments and it's notable they also will be running more of these validation so it would be great to integrate these suggestions for a future submission.

**Justification For Why Not Lower Score:**

NA

**Metareview: Summary, Strengths And Weaknesses:**

This paper propose a deep learning method for predicting functional coding sequences (genes) within genomes. The work is overall well presented and clear to read, and shows state of the art performance comparable to benchmarks. However, it is limited in terms of technique novelty and there's need for more extensive comparison and testing, which would better help to demonstrate the superiority of this method.